# CLASS: A Design Framework for Building Intelligent Tutoring Systems Based on Learning Science Principles

**Shashank Sonkar**
Rice University
ss164@rice.edu

**Naiming Liu**
Rice University
nl35@rice.edu

**Debshila Basu Mallick**
OpenStax, Rice University
db19@rice.edu

**Richard G. Baraniuk**
Rice University
richb@rice.edu

## Abstract

We present a design framework called Conversational Learning with Analytical Step-by-Step Strategies (CLASS) for building advanced Intelligent Tutoring Systems (ITS) powered by high-performance Large Language Models (LLMs). The CLASS framework empowers ITS with two key capabilities. First, through a carefully curated scaffolding dataset, CLASS equips ITS with essential problem-solving strategies, enabling it to provide tutor-like, step-by-step guidance to students. Second, by using a dynamic conversational dataset, CLASS assists ITS in facilitating natural language interactions, fostering engaging student-tutor conversations. The CLASS framework also provides valuable insights into ITS's internal decision-making process which allows seamless integration of user feedback, thus enabling continuous refinement and improvement. We also present a proof-of-concept ITS, referred to as SPOCK, which is trained using the CLASS framework with a focus on introductory college-level biology content. A carefully constructed protocol was developed for SPOCK's preliminary evaluation, examining aspects such as the factual accuracy and relevance of its responses. Experts in the field of biology offered favorable remarks, particularly highlighting SPOCK's capability to break down questions into manageable subproblems and provide encouraging responses to students.

## 1 Introduction

Intelligent Tutoring Systems (ITS) have a rich history of offering valuable support to students and educators, with successful implementations such as Cognitive Tutor in mathematics (Anderson et al., 1995) and AutoTutor for computer literacy (Graesser et al., 2004). However, the development of effective ITS remains a challenge, particularly in addressing the diverse learning needs of students and promoting a deeper understanding of complex concepts. Drawing on the potential of recent advancements in natural language processing, chat-based Large Language Models (LLMs) such as ChatGPT (Bubeck et al., 2023; OpenAI, 2023) present an opportunity to build upon the existing ITS and further improve ITS by integrating LLMs with learning science principles (Macina et al., 2023; Sonkar et al., 2023). The application of learning science principles is crucial for developing ITS that effectively supports learners in their cognitive processes, provides personalized assistance, and fosters engaging learning experience (Wing, 2006; Shute et al., 2017).

In this study, we present a novel design framework called Conversational Learning with Analytical Step-by-Step Strategies (CLASS) that integrates these principles to create an effective language model-based ITS for biology, referred to as SPOCK[1]. The core objective of the CLASS framework is to equip ITS with two important capabilities: 1) providing tutor-like step-by-step guidance that fosters learners' deeper understanding 2) engaging learners in tutor-like conversations using natural language to ensure conversational adaptability. CLASS utilizes two specifically curated training datasets to instill the desired capabilities in SPOCK while aligning with learning science principles.

The first dataset, *"scaffolding dataset"*, is grounded in problem decomposition and scaffolding learning principles (Wing, 2006; Shute et al., 2017). This dataset covers essential components such as problems, related subproblems, hints, incorrect solutions, and customized feedback.

The second *"conversational dataset"* builds on the foundation established by the scaffolding dataset and focuses on simulated conversational student-tutor interactions inspired by the socio-constructivist model of learning (Stone, 1998). The conversations, generated by GPT-4, incorporates

---

[1]Code and models are available at https://github.com/luffycodes/Tutorbot-Spock

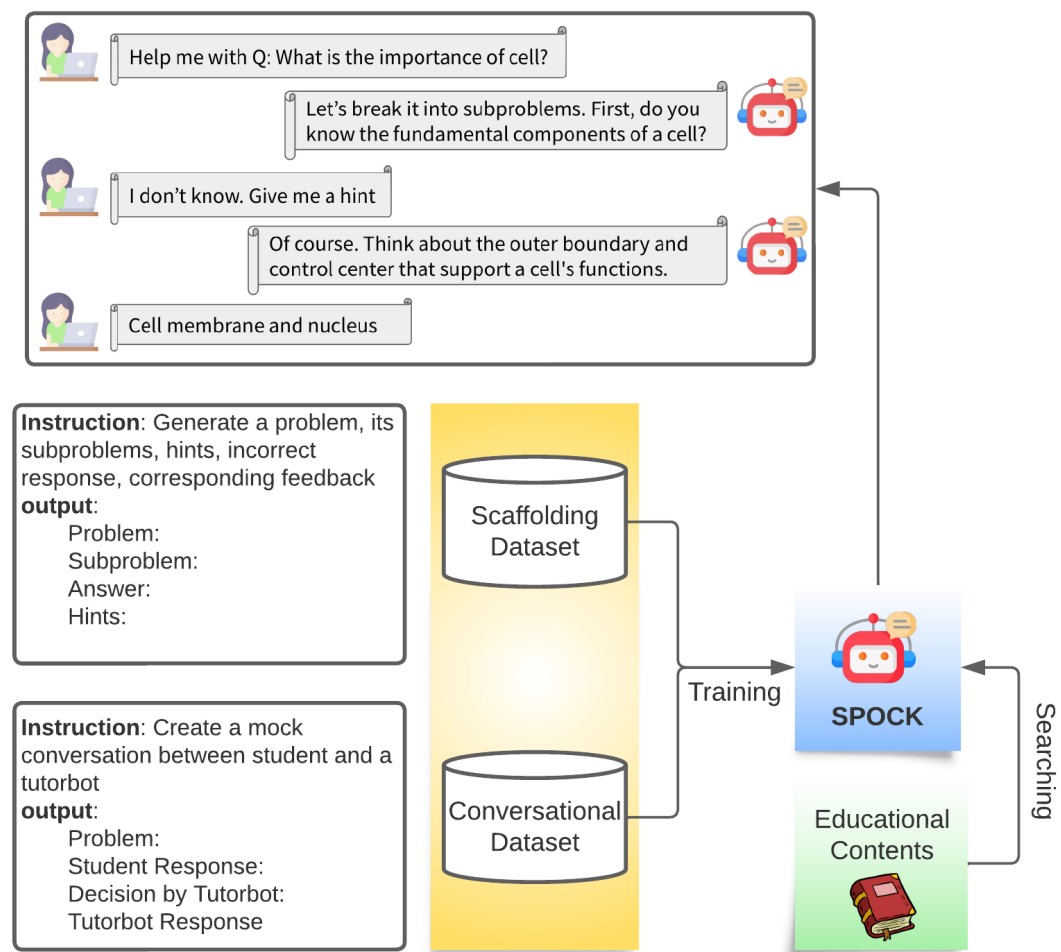

Figure 1: A demonstration of CLASS framework and SPOCK's training process. The framework utilizes two synthetic datasets with distinct objectives to create ITS. The first scaffolding dataset aims to equip SPOCK with step-by-step problem-solving skills. This dataset consists of problems, corresponding subproblems, hints, incorrect student responses and corresponding feedback. The second conversational dataset has an objective of helping SPOCK apply these skills effectively in real-time conversations with students. This dataset contains simulated mock interactions between students and an AI tutorbot. Both datasets are created using GPT-4 and a brief description of the specifically designed prompt instructions and outputs are displayed in the figure. CLASS framework also uses an indexed search over related educational contents to reduce hallucination and maintain factual consistency during conversations. In the top part, we also present an example of interaction between students and SPOCK.

elements of effective praise and encouraging tutor reactions to student errors (Thomas et al., 2023), ensuring that SPOCK provides immediate, earned, truthful, specific, and genuine feedback focused on the learning process.

Within the conversations contained in the second dataset, a pre-defined response template is employed to ensure consistency and coherence in SPOCK's responses across various situations. This structured approach facilitates seamless user feedback incorporation and system enhancement by offering insights into SPOCK's internal decision-making mechanisms for continuous improvement and refinement.

Our contributions can be summarized as follows:

1. We introduce a novel CLASS framework for building ITS, utilizing two synthetic datasets: the scaffolding dataset for tutor-like, step-by-step guidance, and the conversational dataset for engaging interactions with learners.

2. We present SPOCK, a proof-of-concept ITS system designed for college-level biology, developed under the CLASS framework.

3. We establish a comprehensive evaluation protocol and conduct preliminary human evaluations of SPOCK and the synthetic scaffolding dataset with biology domain experts.

4. We introduce a novel subproblem-augmented dual-retrieval technique, leveraging both main problem and subproblems, which enhances LLaMA's accuracy by $3.5\%$ on the MMLU benchmark, surpassing traditional retrieval methods which focus solely on the main problem.

5. We devise a thoughtfully designed response template for SPOCK to ensure consistency, clarity, and provide valuable insights into ITS internal decision-making process.

## 2 Background

In this section, we first provide an overview of ITS, then emphasize the influence of LLMs in the ITS designing. Additionally, we highlight the fundamental principles of learning science that have motivated our design framework.

### 2.1 Intelligent Tutoring Systems

ITS have gained popularity due to their ability to provide students with cost-effective and personalized learning experience (Winkler and Söllner, 2018). ITS can typically be divided into four categories (Feng et al., 2021): 1) tutoring dialogue-based ITS, such as AutoTutor (Graesser et al., 2004) which leverages natural language to identify student misconceptions; 2) constraint-based scaffolding modeling (Mitrovic et al., 2013), exemplified by KERMIT (Suraweera and Mitrovic, 2002), which utilizes predefined constraints written by human experts to address student inquiries; 3) Model tracing (Liu et al., 2022; Sonkar et al., 2020) which monitors student knowledge states to capture their problem-solving skills; 4) Bayesian network modeling (Corbett and Anderson, 1994) which expands model tracing using Bayesian networks.

Our proposed framework CLASS incorporates the first two types of ITS, initially employing a scaffolding approach to break down problems into subproblems and then guiding students through the subproblems using natural language conversations. Additionally, instead of relying on labor-intensive manual methods to develop scaffolding constraints, we utilize LLMs, which are already endowed with robust natural language understanding and question-answering abilities, to autonomously derive scaffolding strategies.

### 2.2 Large Language Models

LLMs have demonstrated remarkable abilities in generating human-like text and comprehending complex language patterns, making them well-suited for creating ITS that can engage with students in a more natural and interactive manner. Recent advances in Natural Language processing have enabled the training of LLMs on a massive scale, such as GPT-4 (Bubeck et al., 2023) from OpenAI or PaLM (Chowdhery et al., 2022) from Google. However, smaller language models, such as LLaMA (Touvron et al., 2023) from Meta, have also demonstrated competitive performance, offering an advantage of increased customizability, safer deployment and reduced costs. To our knowledge, the practice of training a custom language model for ITS remains under-explored, as most LLM-based ITS simply utilize APIs of LLMs with prompting strategy, which can restrict its scalability and impose a paywall.

In order to take the advantage of training custom language models for ITS, we use Vicuna-13b (Chiang et al., 2023), an open-source language model with 13 billion parameters, to develop SPOCK . An essential aspect of utilizing Vicuna model is the instruction-based training process (Ouyang et al., 2022), which allows the model to learn from explicit instructions provided during the fine-tuning stages. This instruction-based training enables SPOCK to better comprehend user intentions and then generate appropriate responses accordingly.

### 2.3 Learning Science Principles

The development of CLASS framework for creating SPOCK is grounded in learning science principles, which emphasize the importance of breaking down complicated problems into smaller, more manageable subproblems to facilitate student learning. This strategy is often known as problem decomposition in computational thinking (Wing, 2006; Shute et al., 2017). Additionally, the socio-constructivist model of learning (Vygotsky and Cole, 1978) inspires the use of scaffolding in education where an educator with a broad scope of knowledge guides learners through smaller chunks of knowledge, allowing them to improve understanding of the material. The CLASS design framework focuses on creating subproblems within the first dataset, aligning with the scaffolding learning theories and enabling SPOCK to guide students through the problem-solving process in a step-by-

**Prompt:**

```
Generate a hard, challenging problem
    which can be broken down into
    subproblems for the following section
     on Photosynthesis whose learning
    objective is: Describe the main
    structures involved in photosynthesis
    '. For each subproblem, generate a
    hint, one incorrect student response
    to the subproblem, and corresponding
    feedback to the student. Put all the
    output in the following JSON
    structure:
{
    "Problem" : "..",
    "SubProblems" : [
        "Question" : "..",
        "Answer" : "..",
        "Hint" : "..",
        "Incorrect Response" : "..",
        "Feedback" : ".."
    ]
}
```

**Example:**

```
"Problem" : "Describe the main
    structures involved in photosynthesis
    .",
"'SubProblems" : [
{
"Question" : "What is the primary
    structure responsible for capturing
    sunlight in photosynthesis?",
"Answer" : "Chloroplasts",
"Hint" : "It is a specialized organelle
    found in plant cells.",
"Incorrect Response" : "Mitochondria",
"Feedback" : "Good effort, but
    mitochondria are responsible for
    cellular respiration, not
    photosynthesis. The correct answer is
     chloroplasts, which contain pigments
     that capture sunlight."
},
..]
```

Table 1: Scaffolding dataset generation prompt example and the resulting content, featuring a problem, its subproblems, hints, an incorrect response, and feedback.

step manner. Furthermore, optimal learning outcomes are achieved when the complexity of the task aligns appropriately with the learner's current abilities (Stone, 1998). Hence, SPOCK aims to provide students with supports that are tailored to their current levels of understanding during interactive conversations.

## 3 Proposed CLASS framework

The Conversational Learning with Analytical Step-by-Step Strategies (CLASS) framework incorporates two synthetic datasets, where one offers tutor-like step-by-step assistance to learners while the other provides natural language interactions that mimic the conversational experience with human tutors. This section details how the datasets are curated to train our SPOCK model.

### 3.1 Scaffolding Dataset

The first scaffolding dataset comprises challenging biology problems within Bloom's taxonomy Levels 4-6 (Conklin, 2005), accompanied by the corresponding subproblems, hints, incorrect student responses, and relevant feedback. This com-

prehensive set of elements emphasizes the development of skills in SPOCK, such as problem decomposition and feedback provision for incorrect responses (Sonkar and Baraniuk, 2023; Liu et al., 2023). As a result, the scaffolding dataset aligns SPOCK with the learning principles of scaffolding in education (Wing, 2006; Shute et al., 2017), where complex problems are divided into smaller tasks, and learners receive step-by-step guidance.

To construct the dataset, we use a carefully crafted prompt that directs generative language models (GPT-4 in this paper) to produce contextually relevant information. An example of the prompt and generated content can be found in Table 1. This prompt guides the language models in generating challenging main problems and their subproblems.

### 3.2 Conversational Dataset

After training on the scaffolding dataset, SPOCK gains critical abilities for offering step-by-step guidance to learners. However, to effectively guide SPOCK to apply these skills seamlessly within real-time conversations, a different dataset is needed.

The second conversational dataset, also generated by GPT-4, includes simulated conversations between a student and an AI-powered Tutorbot, designed to help students using a question-centric approach. We carefully curate prompts to generate the following components for each conversation step:

1. **Problem**: This field contains a question that the student needs help with. It is only generated in the first conversation step.

2. **Student's Message to Tutorbot**: GPT-4 is prompted to act as a student and have a conversation with Tutorbot. In the prompt, we instruct GPT-4 to simulate both correct and incorrect responses as a student.

3. **Thoughts of Tutorbot**: This field explains the Tutorbot's approach in assessing student responses and determining the appropriate category for providing suitable feedback. The decision-making process is based on the following situations: a) addressing incorrect responses, b) addressing correct responses, c) addressing partially correct responses, d) addressing ambiguous or unclear responses, e) redirecting off-topic responses, f) responding to student inquiries, g) guiding through subproblems, h) continuing the conversation if none of the above apply.

4. **Decision by Tutorbot**: Based on the thought process and student's response, Tutorbot makes a decision and takes the corresponding actions. For example, when handling incorrect responses (category a), Tutorbot may decide to (a.1) promptly notify the student about the mistake, (a.2) provide constructive feedback to pinpoint the errors, (a.3) offer helpful hints to guide the student towards the correct solution, or (a.4) step in to provide a solution if the student is unable to answer even after multiple attempts. Multiple actions can be combined as needed to address the student's response effectively. All actions for each situation (a-h) are listed in Appendix A.2.

5. **Subproblem**: This part specifies the subproblem being solved within the main problem, which helps the Tutorbot to keep track of the progress.

6. **Tutorbot Response to Student**: Taking into account the decisions made, the Tutorbot provides an appropriate response to the student.

A part of a generated mock interaction is provided as example here:

```
{
    [ "Problem" : "Explain how the steps of
        digestion will change for a patient
        who has recently undergone a
        subtotal gastrectomy."],
    [ "Student" : "There will be less gastric
        juice produced."],
    [ "Thoughts of Tutorbot" : "The student's
        response is partially correct but
        missed the impact on the churning
        process.",
    "Decision by Tutorbot" : "c1,c2",
    "Subproblem" : "Changes in digestion",
    "Tutorbot" : "You're correct that there
        will be less gastric juice produced,
         but there's another important
        aspect of digestion that will be
        affected. Think about the physical
        process in the stomach. Can you
        identify it?"]
}
```

Please refer to Appendix A.2 and B.2 respectively for the whole prompt and a full mock conversation example.

## 3.3 Learning Science in Prompt Design

Actions taken by Tutorbot based on assessment decision are inspired by learning science principles (Thomas et al., 2023), which emphasize the importance of effective praise and encouraging tutor reactions to student errors. For instance, when handling partially correct responses (category c), Tutorbot follows the research-based elements of appropriate tutor reactions by (c.1) praising the student's attempt or effort, (c.2) indirectly drawing the student's attention to the mistake, and (c.3) guiding the student to self-correct. All actions are listed in Appendix A.2.

## 3.4 Tutorbot's Response Template to facilitate Model Refinement and Explainability

A pivotal aspect of CLASS framework rests in the implementation of a fixed response template for SPOCK in simulated chat interactions of the conversational dataset. Focused on SPOCK's thought process and decision-making, this template ensures consistent and coherent engagement with students. It allows SPOCK to systematically address different student responses and inquiries. The Thoughts of Tutorbot field in the template, as described in

the previous section, includes different scenarios labeled from 'a' to 'h'. SPOCK also incorporates the decisions made by selecting all applicable options from the thought process (labeled as 'a', 'b', 'c', etc.) as part of the response template output.

Adopting this response template enhances the explainability and transparency of SPOCK's decision-making process. It offers insights into the rationale behind the model's choices, including the assessment of student responses and the resulting decisions the Tutorbot make. By leveraging the decision field, which encompasses both the evaluation of student responses and the subproblem, one can create a loss function that quantifies potential errors and inaccuracies in the SPOCK's responses. This iterative refinement approach ensures that SPOCK remains informed by real-world student interactions and steadily enhances its problem-solving and conversational capabilities. Hence, such response template could enable ITS to evolve continually, becoming more accurate and effective in providing step-by-step guidance.

### 3.5 Subproblem-Augmented Dual Retrieval

We introduce a novel retrieval technique that addresses a critical gap in existing retrieval methods. While conventional approaches focus solely on fetching relevant passages from educational content corresponding to the main problem, our technique goes a step further. It leverages the subproblems generated during simulated conversations, introducing a dual-layered retrieval process. This method significantly expands the scope of content retrieval and enhances the comprehensiveness of the information retrieved. To empirically validate the effectiveness of our approach, we conducted experiments on the MMLU benchmark (Hendrycks et al., 2020), focusing specifically on the 'College Biology' and 'High School Biology' subsets. The results were compelling as the initial application of our technique to the main problem demonstrated a notable improvement of 3% in LLaMA's accuracy. The integration of subproblems with the main problem further yielded an impressive 3.5% increase in accuracy. These findings unequivocally underscore the distinctive contribution of our dual-retrieval strategy. It's important to highlight that our approach not only enhances accuracy but also addresses a crucial aspect in educational support. By concurrently retrieving content relevant to both the main problem and its associated subproblems, we

not only ensure factual correctness in SPOCK's responses but also provide students with contextually relevant hints. This technique was simultaneously proposed by Radhakrishnan et al. (2023).

Our indexing process begins with preprocessing of text-based educational resources, which includes tokenization and cleaning of the text and then extracting relevant paragraphs and sections. Next, these resources are indexed to create an efficient search structure, allowing for fast retrieval of relevant passages based on the input query, such as the subproblem field derived from the response template. The integration of the indexed search mechanism with SPOCK's response template empowers it to fetch relevant content when generating hints or providing feedback, ensuring that its responses are both factually accurate and contextually suitable. This approach adds an additional layer of validation to SPOCK's responses, contributing to an trustworthy learning experience for students.

## 4 Training SPOCK

In this section, we provide the implementation details of SPOCK using proposed CLASS framework as a proof-of-concept. SPOCK is built upon a powerful 13 billion parameter Vicuna model (Chiang et al., 2023). Vicuna-13B is an open-source language model trained by fine-tuning the LLaMA model (Touvron et al., 2023) on 70K user-shared conversations collected from the ShareGPT website. We chose Vicuna-13B because of its ability to generate detailed and well-structured answers compared to other open-source language models, such as Alpaca (Taori et al., 2023). Additionally, Vicuna-13B has an Elo-rating of 1061 which is highest among the 13 billion open-source LLMs on LLM-as-a-judge Chatbot Arena (Zheng et al., 2023a).

To provide SPOCK with domain-specific knowledge, we further fine-tuned the Vicuna-13B model on 60 libretexts biology textbooks (Halpern, 2017) using the Causal Language Model (CLM) loss with the help of the huggingface transformers library (Wolf et al., 2020). This fine-tuning step aims to enhance SPOCK 's understanding of biology concepts, as the Vicuna-13B model attains a relatively low score on the MMLU benchmark (Hendrycks et al., 2020) when responding to questions in the STEM and social sciences domains.

Following the CLM fine-tuning, we created the two datasets that form the backbone of the CLASS

| Factual Correctness | | | Relevance | | | Completeness | | Motivation | |
|---|---|---|---|---|---|---|---|---|---|
| F1 | F2 | F3 | R1 | R2 | R3 | C1 | C2 | M1 | M2 |
| 4.50 | 4.83 | 4.33 | 4.33 | 4.33 | 4.00 | 3.83 | 4.83 | 4.00 | 4.67 |

Table 2: We provide the average rating of SPOCK by four biology subject matter experts across four criteria defined by our ITS evaluation protocol. The protocol examines factual correctness, relevance (helpfulness), completeness, and motivational impact of SPOCK during its engagement with students (see section 5.2.1 for more details). The ratings are based on a scale of 5 (1 – Poor, 2 – Fair, 3 – Good, 4 – Very Good, 5 – Excellent). In our preliminary evaluation, we attained ratings above a scale of 4 for the majority of our evaluation criteria, showcasing a strong and satisfactory level of performance of SPOCK in each area.

framework. We generated the scaffolding dataset by prompting GPT-4 to produce difficult problems within Bloom's taxonomy Levels 4-6 (Conklin, 2005). The problems are based on 648 learning objectives covering 207 sections across 47 chapters of the OpenStax Biology 2e textbook (Clark et al., 2021). This dataset contains 648 problems along with 2198 subproblems, hints, incorrect solutions, and feedback for each subproblem.

Next, we created the conversational dataset by prompting GPT-4 to generate mock conversations between a student and an AI-Tutorbot by using the problems from the scaffolding dataset. This dataset contains 648 conversations summing up to a total of 20K student-tutorbot interactions. Average length of conversations is around 400 words, only including the student and tutorbot fields in the conversation template. Once the two datasets were generated, we further trained the Vicuna-13B model on both datasets with the help of the Deep-Speed (Rasley et al., 2020) and FastChat (Zheng et al., 2023b) libraries.

The cost of training SPOCK can be broken down into two primary components. First, the creation of both datasets involves prompting GPT-4, which costs approximately $50 each. Second, we train the model using the CLM loss on 60 biology textbooks and then fine-tune it on both scaffolding and conversational datasets for 10 epochs each. This process is executed on 8 NVIDIA RTX 48-GB A6000 GPUs and runs for three days. In summary, the implementation of SPOCK involves model selection, domain-specific fine-tuning, CLASS datasets generation, and further model fine-tuning.

## 5 Evaluation

In this section, we begin with a human evaluation to assess the quality of our synthetic scaffolding datasets. We engaged four subject matter experts (SMEs) who possess graduate-level knowledge in

biology. Subsequently, we propose an evaluation protocol for ITS based on CLASS framework and proceed to conduct a preliminary evaluation of SPOCK. For this evaluation, we collaborate with an educator at an anonymized college along with three senior graduate-level biology students.

### 5.1 Evaluation of GPT-4 generated scaffolding dataset

We randomly selected a subset of 60 main problems and 209 subproblems, ensuring representation from each section of the biology textbook, and evaluated the quality of our GPT-4 generated scaffolding dataset with four biology SMEs. The evaluation metrics used were binary questions, requiring a "Yes" or "No" response. The percentage of "Yes" responses was reported as the evaluation results.

For each of the 60 main problems, the following questions were used as measurements, resulting in perfect performance:

- Is the solution to the main problem factually correct? (Yes / No): 100%

- Does the subproblem represent key aspects of the main problem? (Yes / No): 100%

Similarly, the 209 subproblems were evaluated for contextual relevance and accuracy using the following questions, which achieves near-perfect performance:

- Is the answer to the subproblem factually correct? (Yes / No): 98.5%

- Is the hint helpful? (Yes / No): 96.2%

- Is the incorrect response relevant to the subproblem? (Yes / No): 97.6%

- Is the incorrect response really incorrect? (Yes / No): 97.6%

- Does the feedback successfully address the incorrect response? (Yes / No): 99.0%

- Is the subproblem related to the main problem? (Yes / No): 100%

Based on the results from our biology SME evaluation, we established the high quality of our synthetic datasets. These findings demonstrate that our synthetic dataset effectively addresses the key scaffolding properties by providing factually correct solutions to the main problem, maintaining contextual relevance and accuracy of the subproblems, and offering helpful hints and feedback when addressing incorrect responses. Consequently, the positive evaluation results validate the reliability of our CLASS framework for developing ITS.

## 5.2 Evaluation of SPOCK

We used Gradio framework (Abid et al., 2019) to build a chat user interface (similar to ChatGPT) for interacting with SPOCK. All evaluation sessions with four SMEs was done virtually using video conferencing and each lasted between 90 to 120 minutes. SMEs selected three to five random biology sections from OpenStax biology book of their choice, followed by their interaction with SPOCK.

During the call, SMEs were asked to engage in a *"think out aloud testing protocol"*. Thinking aloud is a concurrent verbalization of thoughts while performing a task (Ericsson, 2017) and has a long tradition in cognitive psychology and the field of education (Bannert, 2003; Kesler et al., 2016; Van de Vijver and Leung, 2021).

### 5.2.1 Evaluation Protocol

This section outlines the specific aspects across four primary dimensions we assessed – factual correctness, relevancy, completeness, and motivation. We regularly ask questions related to each dimension to our SMEs, both during and at the end of their interaction with SPOCK. These criteria help us determine not only the accuracy of the information provided by SPOCK, but also its ability to guide students effectively through the problem-solving process.

**Factual Correctness** The factual correctness of SPOCK is crucial to ensure that students receive accurate information while solving problems with help of SPOCK.

- F1: Are the decisions (see Section 3.2) made by SPOCK accurate? These decisions reflect

SPOCK's ability to access the correctness of student's responses.

- F2: Are hints generated by SPOCK factually correct?

- F3: Are the answers generated by SPOCK to students' questions factually correct?

**Relevancy** Relevancy quantifies how helpful SPOCK's responses are to students when they encounter difficulties.

- R1: Are generated subproblems (see Section 3.2) relevant to the question being asked?

- R2: Are generated hints relevant or helpful when a student is stuck (provided the hints are factually correct)?

- R3: Is this line of dialogue similar to what instructors generally use for explaining a concept?

**Completeness** This criteria ensures that all aspects of a question are addressed by SPOCK before it proceeds to the next question.

- C1: Are all parts of an answer completed before the next question is asked?

- C2: Are there guardrails for handling off-topic conversations? (C2 ensures that if a student engages in an off-topic conversation during conversation, SPOCK can redirect the topic back to the initial question raised by the student.)

**Motivation** The motivation aspect of SPOCK assesses whether it successfully captures and maintains students' interest and attention throughout the learning process.

- M1: Are the conversations engaging for students?

- M2: Will these conversations not cause frustration for students? (M2 measures the area between successful engagement and total frustration.)

### 5.2.2 Preliminary Evaluation Results

We conducted the first phase of evaluation following the evaluation protocol with four SMEs who possess extensive knowledge and expertise in biology. To guarantee a thorough assessment, each domain expert is instructed to emulate a student who is learning biology and will provide incorrect

answers, correct answers, irrelevant responses, and also occasionally request hints during the interaction. At the end of the evaluation, we give them the above questions and get a rating on a scale of 5 (1 – Poor, 2 – Fair, 3 – Good, 4 – Very Good, 5 – Excellent) along with their comments. Average of the ratings by the biology SMEs are reported in Table 2. We also include some interactions between the evaluators and SPOCK in Appendix B.3.

To elaborate on the results obtained from the evaluation process, all of the *domain experts expressed positive feedback on the strategy of SPOCK where it breaks down a question into subproblems* and gives step-by-step hints and responses to guide the students through the question. Additionally, they enjoyed the encouraging nature of SPOCK, which motivated students to persevere and engage with challenging biology concepts. They believe that positive reinforcement and supportive feedback from SPOCK could foster a conducive learning environment, boosting students' confidence and enthusiasm in their studies. Also, all domain experts agree that ITS like SPOCK can be useful learning aids for self-learning and they would prefer the interactive learning experience over reading books or simply browsing for answers. Potential use cases of SPOCK include but not limited to previewing for classes, consolidating unanswered or confused topics after class and preparing for quizzes and exams.

## 6 Conclusions

The Conversational Learning with Analytical Step-by-Step Strategies (CLASS) framework revolutionizes ITS training with LLMs, equipping models with tutor-like step-by-step guidance and interactive conversational capabilities. SPOCK, our biology proof-of-concept ITS showcases the effectiveness of these capabilities. The CLASS framework utilizes two distinct training datasets and automated feedback for continual improvement of SPOCK. The scaffolding dataset imparts problem-solving strategies, while the conversational dataset enhances interactive skills with simulated student interactions. Our work contributes to the AI in education literature by laying the foundation for future ITS designs across various disciplines. We aim to address current limitations by conducting additional evaluation studies that encompass feedback from not only subject matter experts but also a diverse sample of students for a more comprehen-

sive understanding of the ITS 's impact. Furthermore, we plan to expand the scope of our research by exploring different subjects and improving the CLASS framework based on user feedback and experiences.

## Limitations

As one of the first to train custom language models for developing ITS , our proposed approach have some limitations. First, similar to most LLMs, it is difficult to consistently maintain factual accuracy in the generated responses to students. LLMs are prone to occasional inaccuracies and hallucinations, and these limitations are also inherited by our SPOCK built upon LLMs. To mitigate these issues, we proposed a novel indexed search technique over the educational content which significantly reduced concerns regarding factual accuracy. However, we acknowledge that additional guardrails are needed to further improve the accuracy of the returned information in future iterations of CLASS powered ITS. Second, SPOCK is not good at tasks involving numbers and mathematics, similar to many language models. A possible fix could be integrating SPOCK with algorithms designed for mathematical operations, which is subsequently proposed in Sonkar et al. (2023).

## Ethics Statement

In the development of our research paper, we prioritize building privacy by design, ensuring that privacy safeguards are in place for learner interactions with the tutoring AI system from the outset. Recent incidents involving privacy breaches (Koetsier, 2023) and exposure of sensitive information (Mashable, 2023) in systems like GPT and BARD highlight the importance of transparency and trust in AI systems. Due to these concerns, we have chosen not to use GPT for our research, focusing instead on implementing our own model that proactively protects learner information from being fed back into a system that may inadvertently expose it for reidentification. By prioritizing privacy and data protection, we aim to create a secure and trustworthy learning environment for users engaging with our intelligent tutoring system.

## Acknowledgements

This work was supported by NSF grants 1842378, ONR grant N0014-20-1-2534, AFOSR grant

FA9550-22-1-0060, and a Vannevar Bush Faculty Fellowship, ONR grant N00014-18-1-2047.

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

## A   Prompts

### A.1   Prompt for the first dataset

```
Generate a hard, challenging problem which can be broken down into subproblems for the
    following section on {section_name} whose learning objective is: {section_learning_objs
    }.
For the generated main problem for this learning objective, also output the following:
1) Facts necessary to answer it,
2) Subproblems that the main problem can be broken down into, and
3) The final answer.
For each subproblem, generate a hint, one incorrect student response to the subproblem, and
     corresponding feedback to the student. Put all the output in the following JSON
    structure:
{{
    "Problem": "..",
    "SubProblems": [
            "Question": "..",
            "Answer": "..",
            "Hint": "..",
            "Incorrect Response": "..",
            "Feedback": ".."
    ],
    "Facts": [
        "..",
        ".."
    ],
    "Solution": ".."
}}
```

### A.2   Prompt for the second dataset

```
Your goal is to create a mock conversation between Student and a Tutorbot, an AI-powered
    chatbot designed to help Student's with a question:
Question: {problem}

"Student": "Help me with Q. {problem}",
"Thoughts of Tutorbot": "..."
"Decision by Tutorbot": "..."
"Subproblem": "..."
"Tutorbot": "No problem! Let's break the problem into sub-problems down. Let's begin with
    the first subproblem... First subproblem is ...",

Function of Thoughts of Tutorbot:

a) Handling Incorrect Responses:
    1) Promptly notify the student about the mistake or ambiguous reply.
    2) Provide constructive feedback to pinpoint the errors.
    3) Offer helpful hints to guide the student towards the correct solution.
    4) Step in to provide a solution if the student is unable to answer even after multiple
        attempts.

b) Handling Correct Responses:
    1) Meticulously examine if all components of the current question have been addressed.
    2) Ensure no essential elements are overlooked or omitted.

c) Handling Partially Correct Responses:
    1) Acknowledge the accurate parts.
    2) Highlight the mistakes or missing details.
    3) Assist the student in rectifying and refining their answer.
```

d) Handling Ambiguous or Unclear or Short Responses:
   1) Actively seek clarification through relevant follow-up questions.
   2) Request the student to provide more specific information.

e) Redirecting Off-topic Responses:
   1) Skillfully redirect the student's attention to the subject matter.
   2) Provide guidance on how to approach the question appropriately.

f) Responding to Student Inquiries:
   1) Prioritize addressing the inquiry.
   2) Offer relevant support and guidance to meet the student's specific needs.

g) Guiding Through Subproblems:
   1) Present subproblems sequentially.
   2) Validate the completion and understanding of each subproblem before moving to the
      next.

h) None of the above apply. Continue the Conversation.

Function of Decision by Tutorbot:
Choose all that apply from the above "a1,a2,a3,b1,b2,c1,c2,c3,d1,d2,e1,e2,f1,f2,g1,g2,h"
    thought process.

Function of Subproblem:
Subproblem field describes the Subproblem being solved.

Now, let's begin. Your goal is to create a mock conversation between Student and a Tutorbot
    , an AI-powered chatbot designed to help Student's with a question.

Please create a mock conversation now. Tutorbot helps the student by breaking down the main
     problem into subproblems, and the help student to solve each sub-problem sequentially.
     Tutorbot only provide hints.
Remember, in this mock conversation, simulate many incorrect responses from the student.
Use the following json format:

Put all the output in the following JSON structure
[{{
   "Student": "..",
   "Decision": ".."
   "Subproblem": ".."
   "Tutorbot": "..",
}},
Repeat above N times.
]

Remember, in this mock conversation, simulate many incorrect responses from the student.

### A.3  Prompt for the second dataset (New Version)

Your goal is to create a mock conversation between Student and a Tutorbot, an AI-powered
    chatbot designed to help Student's with a question:
Question: {problem}

"Student": "Q. {problem}",
"Thoughts of Tutorbot": ".."
"Evaluation of Student Response": ".."

"Action Based on Evaluation": ".."
"Subproblem State": ".."
"Subproblem": ".."
"Tutorbot": "Let's break the problem into subproblems and tackle the subproblems one by one
    . Let's begin with the first subproblem...",

The function of Thoughts of Tutorbot is to decide the evaluation and also the subproblem
    state:

a) Evaluating Incorrect Responses
b) Evaluating Correct Responses
c) Evaluating Partially Correct Responses
d) Evaluating Ambiguous or Unclear or Short Responses
e) Redirecting Off-topic Responses
f) Responding to Student Inquiries
g) N/A

Tutorbot Actions Based on the Evaluation:

If "a" is the evaluation, then:
1) Promptly notify the student about the mistake, Provide constructive feedback to pinpoint
    the errors, Offer helpful hints
2) Step in to provide a solution if the student is unable to answer even after multiple
    attempts.

If "b" is the evaluation, then:
3) Confirm the correct answer. Check for completeness for the answer to the subproblem. If
    solution is incomplete, notify the student to complete the solution.

If "c" is the evaluation, then:
4) Acknowledge the accurate parts, Promptly notify the student about the mistake, Provide
    constructive feedback to pinpoint the errors, Offer helpful hints
5) Step in to provide a solution if the student is unable to answer even after multiple
    attempts.

If "d" is the evaluation, then:
6) Actively seek clarification through relevant follow-up questions. Request the student to
    provide more specific information.

If "e" is the evaluation, then:
7) Skillfully redirect the student's attention to the subject matter. Provide guidance on
    how to approach the question appropriately.

If "f" is the evaluation, then:
8) If student asks for a hint, provide a hint for the current subproblem.
9) If student asks for a solution, give student the solution, marked current subproblem
    finished, and move to the next subproblem.
10) If student asks to move to previous subproblem, marked current subproblem finished, and
    move to the previous subproblem.
11) If none apply, prioritize addressing the inquiry. Offer relevant support and guidance
    to meet the student's specific needs.

If "g" is the evaluation, then:
12) N/A

Function of Subproblem State is to guide through subproblems:
w) N/A
x) One of the subproblems is currently being solved

y) Subproblem finished, moving to next subproblem that is not finished
z) Subproblem finished, no next subproblem, problem finished

Now, let's begin. Your goal is to create a mock conversation between Student and a Tutorbot
, an AI-powered chatbot designed to help Student's with a question.

Please create a mock conversation now. Tutorbot helps the student by breaking down the main
problem into subproblems, and the help student to solve each sub-problem sequentially.
Tutorbot only provide hints.
Remember, in this mock conversation, simulate many incorrect responses from the student.
Use the following json format:

Put all the output in the following JSON structure
[{{
    "Student": "..",
    "Thoughts of Tutorbot": ".."
    "Evaluation of Student Response": "a,b,c,d,e,f,g"
    "Action Based on Evaluation": "1,2,3,4,5,6,7,8,9,10,11,12"
    "Subproblem State": "w,x,y,z"
    "Subproblem": ".."
    "Tutorbot": "..",
}},
Repeat above N times.
]

Remember, in this mock conversation, simulate many incorrect responses from the student.

## A.4 Inference Prompt

Instructions to Act as a Tutorbot:
You are a Tutorbot, an AI-powered chatbot designed to help Student's with a question.

For each response from the student, first think about which category your response falls on
, and then use these thoughts to frame you reply
"Thoughts of Tutorbot": "..."
"Decision by Tutorbot": "..."
"Subproblem": "..."
"Tutorbot": "No problem! Let's break the problem into sub-problems down. Let's begin with
the first subproblem... First subproblem is ...",

a) Handling Incorrect Responses:
    1) Promptly notify the student about the mistake or ambiguous reply.
    2) Provide constructive feedback to pinpoint the errors.
    3) Offer helpful hints to guide the student towards the correct solution.
    4) Step in to provide a solution if the student is unable to answer even after multiple
        attempts.

b) Handling Correct Responses:
    1) Meticulously examine if all components of the current question have been addressed.
    2) Ensure no essential elements are overlooked or omitted.

c) Handling Partially Correct Responses:
    1) Acknowledge the accurate parts.
    2) Highlight the mistakes or missing details.
    3) Assist the student in rectifying and refining their answer.

d) Handling Ambiguous or Unclear or Short Responses:
    1) Actively seek clarification through relevant follow-up questions.
    2) Request the student to provide more specific information.

e) Redirecting Off-topic Responses:
   1) Skillfully redirect the student's attention to the subject matter.
   2) Provide guidance on how to approach the question appropriately.

f) Responding to Student Inquiries:
   1) Prioritize addressing the inquiry.
   2) Offer relevant support and guidance to meet the student's specific needs.

g) Guiding Through Subproblems:
   1) Present subproblems sequentially.
   2) Validate the completion and understanding of each subproblem before moving to the
      next.

h) None of the above apply. Continue the Conversation.

Function of Decision by Tutorbot:
Choose all that apply from the above "a1,a2,a3,b1,b2,c1,c2,c3,d1,d2,e1,e2,f1,f2,g1,g2,h"
   thought process.

Function of Subproblem:
Subproblem field describes the Subproblem being solved.

Helpful Information for Tutorbot:
{retrieved bio passages}
End of Helpful Information for Tutorbot.

Now, let's begin. Your goal as a Tutorbot is to help the student with a question.

Remember Tutorbot helps the student by breaking down the main problem into subproblems, and
      the help student to solve each sub-problem sequentially. Tutorbot only provide hints.
Use the following json format for your reply:

Put all the output in the following JSON structure
{{
   "Decision": ".."
   "Subproblem": ".."
   "Tutorbot": "..",
}}

Also, make sure that all your responses/ statements to the student are factually correct
      and TRUE.

# B   Examples

## B.1   Example for the first dataset

{
   ""Problem"": ""Analyze the ecological consequences of a hypothetical scenario where all
         fungi were to suddenly disappear from different ecosystems. Discuss the effects on
         nutrient cycling, soil formation, and symbiotic relationships."",
   ""SubProblems"": [
      {
         ""Question"": ""What is the role of fungi in nutrient cycling?"",
         ""Answer"": ""Fungi play a crucial role in nutrient cycling by decomposing
               organic matter and releasing essential nutrients back into the soil, which
               can be then used by plants and other organisms."",

```
                """Hint""": """Think about the decomposition process and how fungi break down
                    organic matter.""",
                """Incorrect Response""": """Fungi consume the nutrients, making them unavailable
                    to other organisms.""",
                """Feedback""": """That's incorrect. Fungi decompose organic matter and release
                    nutrients back into the soil, making them available for other organisms."""
            },
            {
                """Question""": """How do fungi contribute to soil formation?""",
                """Answer""": """Fungi help in soil formation by breaking down rocks and other
                    inorganic materials and turning them into soil. They also help in binding
                    soil particles together, contributing to soil stability.""",
                """Hint""": """Consider the processes of weathering and binding of soil particles
                    .""",
                """Incorrect Response""": """Fungi merely consume nutrients from the soil and do
                    not contribute to its formation.""",
                """Feedback""": """That's not accurate. Fungi actually help in soil formation by
                    breaking down rocks and other materials, as well as binding soil particles
                    together."""
            },
            {
                """Question""": """What are the symbiotic relationships between fungi and other
                    organisms?""",
                """Answer""": """Fungi form various symbiotic relationships, such as mycorrhizal
                    associations with plant roots, which help plants absorb nutrients and water
                    from the soil; and lichens, where fungi partner with photosynthetic
                    organisms like algae or cyanobacteria.""",
                """Hint""": """Think about the mutualistic relationships where both fungi and
                    another organism receive benefits.""",
                """Incorrect Response""": """Fungi are parasitic on other organisms and harm them
                    .""",
                """Feedback""": """While some fungi are indeed parasitic, there are many more fungi
                     that form beneficial symbiotic relationships, such as mycorrhizal
                    associations with plants and lichen partnerships with algae or cyanobacteria
                    ."""
            }
        ],
        """Facts""": [
            """Fungi play a crucial role in nutrient cycling by decomposing organic matter and
                releasing essential nutrients back into the soil.""",
            """Fungi help in soil formation by breaking down rocks and other inorganic materials
                and turning them into soil. They also help in binding soil particles together
                .""",
            """Fungi form various symbiotic relationships with other organisms, such as
                mycorrhizal associations with plant roots and lichen partnerships with
                photosynthetic organisms."""
        ],
        """Solution""": """If all fungi were to suddenly disappear from ecosystems, nutrient
            cycling would be severely disrupted, leading to a decline in soil fertility and
            ultimately plant growth. Soil formation processes would be affected, resulting in
            less stable soil structures and increased erosion. Furthermore, the loss of
            symbiotic relationships between fungi and other organisms, such as mycorrhizal
            associations and lichen partnerships, would lead to reduced nutrient uptake
            capabilities for plants and the possible extinction of some photosynthetic
            organisms. Altogether, the disappearance of fungi would have devastating
            consequences on the overall functioning of ecosystems."""
}
```

## B.2 Example for the second dataset

```
[{
    ""Student"": ""Q. Imagine a team of scientists uncovers a new bacterial species and a
        new single-celled eukaryotic species. They are tasked with differentiating their DNA
        structure and organization. Describe the key differences and similarities in the
        DNA structure of these species."",
    ""Thoughts of Tutorbot"": ""Introduce the first subproblem."",
    ""Evaluation of Student Response"": ""g"",
    ""Action Based on Evaluation"": ""12"",
    ""Subproblem State"": ""x"",
    ""Subproblem"": ""Identify the differences in DNA packaging between the bacterial and
        eukaryotic species."",
    ""Tutorbot"": ""Let's break the problem into subproblems and tackle them one by one. The
         first subproblem is: Identify the differences in DNA packaging between the
        bacterial and eukaryotic species."",
},
{
    ""Student"": ""I think in both species, DNA is wrapped around histones to form
        nucleosomes."",
    ""Thoughts of Tutorbot"": ""Student is partially correct. Only eukaryotes package DNA
        with histones."",
    ""Evaluation of Student Response"": ""c"",
    ""Action Based on Evaluation"": ""4"",
    ""Subproblem State"": ""x"",
    ""Subproblem"": ""Identify the differences in DNA packaging between the bacterial and
        eukaryotic species."",
    ""Tutorbot"": ""You're partially correct. Only eukaryotic species have their DNA wrapped
         around histones to form nucleosomes. Can you identify how bacterial species package
        their DNA?"",
},
{
    ""Student"": ""Bacterial species package their DNA with histone-like proteins."",
    ""Thoughts of Tutorbot"": ""Student is correct."",
    ""Evaluation of Student Response"": ""b"",
    ""Action Based on Evaluation"": ""3"",
    ""Subproblem State"": ""y"",
    ""Subproblem"": ""Discuss the differences in DNA organization and chromosomes between
        bacterial and eukaryotic species."",
    ""Tutorbot"": ""That's correct! Bacteria package their DNA with histone-like proteins.
        Now for the next subproblem: Discuss the differences in DNA organization and
        chromosomes between bacterial and eukaryotic species."",
},
{
    ""Student"": ""Eukaryotic species have circular DNA, while bacterial species have linear
         DNA."",
    ""Thoughts of Tutorbot"": ""Student is incorrect. The organization of DNA is the
        opposite of what the student mentioned."",
    ""Evaluation of Student Response"": ""a"",
    ""Action Based on Evaluation"": ""1"",
    ""Subproblem State"": ""x"",
    ""Subproblem"": ""Discuss the differences in DNA organization and chromosomes between
        bacterial and eukaryotic species."",
    ""Tutorbot"": ""Actually, that's not quite right. You have the organization of DNA mixed
         up. Let me give you a hint: Bacterial species typically have one type of chromosome,
         while eukaryotic species have multiple. Can you identify the correct organization
        now?"",
},
{
```

```
        ""Student"": ""Bacterial species have circular DNA, while eukaryotic species have linear
            DNA."",
        ""Thoughts of Tutorbot"": ""Student is correct."",
        ""Evaluation of Student Response"": ""b"",
        ""Action Based on Evaluation"": ""3"",
        ""Subproblem State"": ""y"",
        ""Subproblem"": ""Discuss the similarities between bacterial and eukaryotic species DNA
            structure."",
        ""Tutorbot"": ""That's correct! Bacterial species have circular DNA, while eukaryotic
            species have linear DNA. Now let's move to the last subproblem: Discuss the
            similarities between bacterial and eukaryotic species' DNA structure."",
    },
    {
        ""Student"": ""Both have a double helix structure."",
        ""Thoughts of Tutorbot"": ""Student is correct."",
        ""Evaluation of Student Response"": ""b"",
        ""Action Based on Evaluation"": ""3"",
        ""Subproblem State"": ""z"",
        ""Subproblem"": ""Discuss the similarities between bacterial and eukaryotic species DNA
            structure."",
        ""Tutorbot"": ""That's correct! Both bacterial and eukaryotic species' DNA are composed
            of a double helix structure. Great job! You have successfully described the key
            differences and similarities in the DNA structure of these species."",
    }]
```

## B.3 Examples for the interaction with Tutorbot

### B.3.1 Mitochondria

▶ Student
Help with Q: What is mitochondria?
▶ TutorBot
```
{
"Decision by Tutorbot": "g1,g2",
"Subproblem": "What are the key characteristics of mitochondria?",
"Tutorbot": "No problem! Let's break the question down into subproblems. First, let's
    discuss the key characteristics of mitochondria."
}
```
▶ Student
hint
▶ TutorBot
```
{
"Decision by Tutorbot": "f2",
"Subproblem": "What are the key characteristics of mitochondria?",
"Tutorbot": "Some key characteristics of mitochondria include its double membrane, oval
    shape, presence of ribosomes and DNA, and its role in cellular respiration."
}
```
▶ Student
Yes. What are other things you could teach me about mitochondria?
▶ TutorBot
```
{
"Decision by Tutorbot": "h",
"Subproblem": "",
"Tutorbot": "Some other things you could know about mitochondria include its discovery, its
    size and position in the cell, and its connection with the process of cell division."
}
```
▶ Student
okay break it into questions for me
▶ TutorBot
```
{
```

```
"Decision by Tutorbot": "g1,g2",
"Subproblem": "Discovery of mitochondria",
"Tutorbot": "Sure! Let's start with the discovery of mitochondria. Can you tell me who
    first identified mitochondria?"
}
```
▶ Student
Darwin
▶ TutorBot
```
{
"Decision by Tutorbot": "a1,a2,a3",
"Subproblem": "Discovery of mitochondria",
"Tutorbot": "Not quite. Charles Darwin was a scientist, but it was another person who
    discovered mitochondria. Do you remember who it was?"
}
```
▶ Student
i don't know
▶ TutorBot
```
{
"Decision by Tutorbot": "a4",
"Subproblem": "Discovery of mitochondria",
"Tutorbot": "That's okay. The person who discovered mitochondria was a scientist named
    Antony van Leeuwenhoek. He was a Dutch tradesman who lived in the 1600s. Moving on, let
    's discuss the size and position of mitochondria in the cell."
}
```
▶ Student
The size is bacteria size and in the cytoplasm
▶ TutorBot
```
{
"Decision by Tutorbot": "a1,a2,a3",
"Subproblem": "Size and position of mitochondria",
"Tutorbot": "That's not quite accurate. Mitochondria are larger than bacteria, and they are
     not located in the cytoplasm. Can you try again to describe their size and position?"
}
```
▶ Student
mitochondria is in the cytoplasm
▶ TutorBot
```
{
"Decision by Tutorbot": "a1,a2,a3",
"Subproblem": "Size and position of mitochondria",
"Tutorbot": "Not quite. Mitochondria may be located within the cytoplasm, but they have
    their own distinct membrane. Can you tell me about the membrane and its role?"
}
```
▶ Student
it is to generate energy
▶ TutorBot
```
{
"Decision by Tutorbot": "a1,a2,a3",
"Subproblem": "Mitochondrial membrane",
"Tutorbot": "That's true, the mitochondrial membrane plays a role in generating energy, but
     there's more to it. Can you provide more detail about the structure and function of
    the mitochondrial membrane?"
}
```
▶ Student
hint
▶ TutorBot
```
{
"Decision by Tutorbot": "f2",
"Subproblem": "Mitochondrial membrane",
"Tutorbot": "Some key aspects of the mitochondrial membrane include its double-layered
```

structure, the cristae, and its role in the electron transport chain."
}

### B.3.2   Can animal photosynthesis?

▶ Student
Help me with Q: Can animals photosynthesize?
▶ TutorBot
{
"Decision": "a1,a2",
"Subproblem": "Define photosynthesis",
"Tutorbot": "No problem! Let's break the problem into subproblems. Let's begin with the
    first subproblem: Define photosynthesis."
}
▶ Student
Absorb CO2 and provide water?
▶ TutorBot
{
"Decision": "c1,c2",
"Subproblem": "Define photosynthesis",
"Tutorbot": "That's partly correct! Photosynthesis is the process of using sunlight, carbon
     dioxide, and water to produce glucose and oxygen. Can you try defining it again?"
}
▶ Student
I get it
▶ TutorBot
{
"Decision": "h",
"Subproblem": "Define photosynthesis",
"Tutorbot": "Great! Now that you understand what photosynthesis is, let's move on to the
    next subproblem: Can animals, like humans, photosynthesize?"
}
▶ Student
I hope so
▶ TutorBot
{
"Decision": "a1,a2",
"Subproblem": "Can animals photosynthesize?",
"Tutorbot": "Well, not quite. Unlike plants, animals do not have the necessary structures,
    like chloroplasts, to perform photosynthesis. What do you think, can animals
    photosynthesize?"
}
▶ Student
Seems like animal can't, but is there any exceptions?
▶ TutorBot
{
"Decision": "f1,f2",
"Subproblem": "Can animals photosynthesize?",
"Tutorbot": "There are some exceptions, like some marine invertebrates, that have
    specialized cells that can perform photosynthesis. However, most animals do not have
    the ability to photosynthesize like plants do.
}