# OpenReview forum: "CLASS: A Design Framework for Building Intelligent Tutoring Systems Based on Learning Science principles"
_EMNLP/2023/Conference — EMNLP 2023 Findings_

### Official Review · Reviewer_g1rc · 2023-08-02

**Soundness:** 3

**Excitement:**

4: Strong: This paper deepens the understanding of some phenomenon or lowers the barriers to an existing research direction.

**Paper Topic And Main Contributions:**

Aiming to design Intelligent Tutoring System (ITS) system that are based on Learning Science Principals, this paper
(1) Proposed a framework called CLASS
(2) Trained a proof of concept ITS called Spock in the domain of teaching student to learn biology concept / taxonomy
(3) Defined an evaluation protocol
(4) Provided a more qualitative evaluation of the built system with 4 experts showing the positive outcome

Based on what they did, I believe the contributions:
1. The framework to allow building ITS that can break hard / bigger concept into smaller and more solvable problems itself with Scaffolding and conversational dataset is interesting is novel and needed
2. Using LLMs to generate both scaffolding and conversation datasets for ITS is a scalable approach
3. The approach to training Spock and building Spock can be a good example for others to learn how to build LLM based ITS / Chatbot

**Questions For The Authors:**

1. The prompt engineering. This paper heavily relies on prompt engineering to find the curated prompt to generate the right results. It would be great if the author can share briefly how it was done and why they chose the one used in the paper.
2. Generated content accuracy. The scalability of the approach comes from the LLM generation. It would be great if the author can share if some accuracy numbers can be reported, it doesn't require examine all of the generate questions, the accuracy based on a sample will be beneficial
3. How student's responses is evaluated. I couldn't find the details about how the system determine is student's response is right / wrong / partially correct. I believe this is a critical component for the ITS. It would be great if the authors can emphasize it.

**Reasons To Accept:**

1. Articulated an interesting interaction of LLM, ITS, and Learning Principle. Proposed a framework, built an example system in a specific domain, achieved positive results. Not all of single contribution is new, however, combining them together provides a new, scalable (and working) framework to build more personal ITS that is based on the learning principals
2. The approach of using GPT-4 to generate datasets, then using datasets to fine-tune a LLAMA based LLM for a specific use case provided an example use case of how to combining multiple LLM to build NLP based application.

**Reasons To Reject:**

1. The evaluation results are still subjective even though there are numeric scores. Even the experts gave positive compliments, the users / students feedback is missing. It's hard to see if the designed system really helped students learn what they intended to learn. Some users test or experiments would make this paper much stronger to show the value of proposed framework and approach.
2. Clarity of the doc: There are multiple things mentioned in the paper: CLASS, Spock, content index. It's not super clear to me what they include. Is CLASS just mean two datasets? Is Spock a system or just the LLM tuned? How is the content index integrated into the system? It might be worth it to add a diagram about Spock to show what's inside Spock, even better if we can see a sequence diagram. That will increase readers' understanding of how it works
3. Some other question below:




**Reproducibility:**

3: Could reproduce the results with some difficulty. The settings of parameters are underspecified or subjectively determined; the training/evaluation data are not widely available.

**Reviewer Confidence:**

4: Quite sure. I tried to check the important points carefully. It's unlikely, though conceivable, that I missed something that should affect my ratings.

---

> ### Author Rebuttal · Authors · 2023-08-29
>
> We thank you for your thoughtful feedback. You summarized our contributions eloquently: 1) CLASS framework including scaffolding and conversational datasets, 2) Proof-of-concept Biology LLM-based ITS, and 3) evaluation protocol for ITS along with domain expert evaluation. Below, we address your concerns and answer your questions.
>
> *Comment # 1: Evaluation results are subjective.*
>
> We included preliminary evaluations by four subject matter experts (SMEs) which included one lecturer with many years of college teaching experience and three senior biology PhD graduate students with extensive grading experience. One of the main feedback points we sought in the iterative prompt design process was whether the ITS was perceived to be pedagogically useful to the students. We found that the SMEs concurred that the SPOCK ITS would be useful for learning biology. We agree that evaluation with real students is critical for the design of any ITS, and we have planned to start our student evaluation this fall semester.
>
> *Comment #2: Clarity of the paper.*
>
> CLASS refers to the design framework that provides guidelines to create synthetic scaffolding and conversational datasets, and subsequently uses these datasets to train an open source LLM-based ITS.
>
> SPOCK is a proof-of-concept Biology ITS developed using the CLASS framework with Vicuna-13B as the base model and GPT-4 generated datasets as the training datasets.
>
> The content index is implemented using the two steps: 1). Embedding the passages in open-source OpenStax biology textbook using the sentence-transformers library and sentence-t5-xxl model. 2). At each step of the conversation, we parse the main question posed by the student at the beginning of the conversation and the subproblem field in tutorbot response to retrieve relevant passages from the book and we append this information at the beginning of the conversation. This information can be used for the generation of the next response. We will duly modify the diagram to help the readers understand the process better.
>
> *Question #1: How and why are the prompts selected?*
>
> The design of the prompt was **an iterative process with continuous feedback from SMEs**. We will provide some explanations on how / why we developed our prompts (L909-1000). 1).  One of the suggestions of the SMEs was to understand the rationale behind the tutorbot response. Hence, we introduce the “Thoughts of the Tutorbot” field (L990) to better understand why the Tutorbot provides the responses. 2). We include the “Subproblem” field (L994) to help us retrieve the right contents needed by LLM to correctly answer/address student’s queries. 3). With the help of SMEs, we come up with exhaustive and nearly mutually exclusive evaluation (L923-932) and action (L934-970) fields to guide the Tutorbot’s responses. Our assumption is that the generation of accurate evaluation and action will lead to better Tutorbot responses since the final response generation is conditioned on evaluation and action outcomes. This design could enable us to easily collect feedback since we could evaluate the selection of the evaluation and action.
>
> *Question #2: Assessment of the GPT-4 generated datasets:*
>
> We thank you for this feedback that helps us strengthen our work. We evaluated the quality of the GPT-4 generated dataset with four biology subject matter experts (SMEs). We selected a subset of 60 main problems and 209 subproblems (an example can be found in Table 1) and used the questions provided below as the evaluation metrics. We present the percentage of “Yes” as the evaluation results below. From the evaluation results from our biology SMEs, we are confident that our synthetic datasets are of high quality. We will include these details in the paper if accepted and hope this could address your concerns about dataset quality.
>
> - For each main problem (60 in total), we used the following questions as the measurements and received perfect performance:
>     - Is the solution to the main problem factually correct? (Yes / No): 100%
>     -  Does the subproblem represent key aspects of the main problem? (Yes / No): 100%
> - The evaluation also established that the subproblems (209 in total)  were contextually relevant and accurate based on the following questions:
>     - Is the answer to the subproblem factually correct? (Yes / No): 98.5%
>     - Is the hint helpful? (Yes / No): 96.2%
>     - Is the incorrect response relevant to the subproblem? (Yes / No): 97.6%
>     - Is the incorrect response really incorrect? (Yes / No): 97.6%
>     - Does the feedback successfully address the incorrect response? (Yes / No): 99.0%
>     - Is the subproblem related to the main problem? (Yes / No): 100%
>
> *Question #3: How does the model evaluate student responses during generation?*
>
> The evaluation/action based on the student response is also generated by the LLM. Subsequently, conditioned on the evaluation and action pair, the final response is generated.
> For each student response, tutorbot outputs the following json (L932-L979):\
> {\
>    "Thoughts of Tutorbot": "..",\
>    "Evaluation of Student Response": "a,b,c,d,e,f,g",\
>    "Action Based on Evaluation": "1,2,3,4,5,6,7,8,9,10,11,12",\
>    "Subproblem State": "w,x,y,z",\
>    "Subproblem": "..",\
>    "Tutorbot": "..",\
> }\
> Our conversational dataset is also generated in this format. We use Vicuna-13B as our base model; fine-tuning Vicuna on the scaffolding and conversational dataset enhances its ability to accurately classify the correctness of student answers during the generation process.
>
> *Reproducibility*
>
> The code with detailed instructions and links to downloading open-source datasets and models can be anonymously accessed here: https://anonymous.4open.science/r/Tutorbot-Spock-FB4F.

---

### Official Review · Reviewer_c1QD · 2023-08-02

**Soundness:** 3

**Excitement:**

3: Ambivalent: It has merits (e.g., it reports state-of-the-art results, the idea is nice), but there are key weaknesses (e.g., it describes incremental work), and it can significantly benefit from another round of revision. However, I won't object to accepting it if my co-reviewers champion it.

**Paper Topic And Main Contributions:**

The authors introduce an Intelligent Tutoring System framework called CLASS, which provides two capabilities: a scaffolding dataset for walking students through problems, and a conversational dataset for communicate with students. Moreover, the authors train SPOCK, a large language model designed to interact with students. Finally, the authors provide a preliminary evaluation of SPOCK within the CLASS framework assessed by various biology experts.

**Questions For The Authors:**

A)	General: Will the datasets (scaffolding and conversational) be released?

B)	Will the model (Vicuna-13B) be released?

C)	Lines 046-047: This seems to insinuate that existing ITS systems do not take into account “learning science principles”. I am not intimately familiar with ITS systems, but I find it hard to believe that this is the case. Do you have any citation or analysis for this claim?

D)	Lines 141-143: Claiming that LLMs are “already endowed with robust natural language understanding and strong reasoning capabilities” is a bit general, and I think that many practitioners in the field would disagree with this claim. Perhaps you can make it more specific (e.g., LLMs have strong conversational or question-answering abilities)?

E)	Lines 225-227: The wording “like GPT-4” implies that you use other language models to generate this dataset? Or it is just GPT-4?

F)	Lines 369-370 and lines 384-386: Do you have any analysis showing that retrieving passages before generation ensures SPOCK’s correctness? From my experience with retrieval-augmented generation (RAG), the model may still deviate from provided input content.

G)	Lines 457-459: Why wasn’t the evaluation done with students, instead of biology SMEs?

**Reasons To Accept:**

The strengths of this paper are as follows:

-	The constructions of a scaffolding and conversational dataset for ITS.

-	A framework (CLASS) for training LLMs on ITS-based tasks using the aforementioned datasets.

-	A preliminary evaluation of the feasibility of using such a framework + trained model in the biology domain.

**Reasons To Reject:**

The weaknesses of this paper are as follows:

-	There was no assessment of the constructed datasets, neither the scaffolding nor the conversational. With both datasets were built by GPT-4, and with GPT-4 simulates both sides of the conversation in the conversational dataset (lines 238-240), I found it difficult to determine the quality of the datasets from the paper. I would suggest performing some brief analysis on the quality of the generated datasets (e.g., some sampled human evaluation).

-	The framework dataset construction process was only performed with GPT-4. With GPT-4 being closed and commercial, many instructors may be interested in leveraging open-source models which avoid sending data to third parties. Testing the framework construction process on models beyond GPT-4 would (1) show the generality of the framework, and (2) provide a concrete, reproducible method for educators to build without using 3rd party models.

-	There were no baselines to compare SPOCK against. SPOCK was trained using a single model (Vicuna-13B), which was pre-trained using biology textbooks. It is unclear (1) how the model would have done without the CLASS framework dataset fine-tuning, (2) how the textbook pre-training affected performance, and (3) how other models would compare to Vicuna-13B). I’d suggest presenting the performance of Vicuna-13B with and without (1), (2), and compared to a few other LLMs.

**Reproducibility:**

3: Could reproduce the results with some difficulty. The settings of parameters are underspecified or subjectively determined; the training/evaluation data are not widely available.

**Reviewer Confidence:**

3: Pretty sure, but there's a chance I missed something. Although I have a good feel for this area in general, I did not carefully check the paper's details, e.g., the math, experimental design, or novelty.

**Typos Grammar Style And Presentation Improvements:**

-	Line 233: The phrasing “After training on the…” to begin the section was a jump that I wasn’t expecting, as the previous section had no mention of training anything.

-	Line 711: The URL is overflowing in the reference.

---

> ### Author Rebuttal · Authors · 2023-08-29
>
> We are grateful for your review and valuable feedback. We are glad that you appreciated our design framework to train an LLM-based ITS with scaffolding properties.
>
> *Comment #1: Assessment of the GPT-4 synthetic training datasets*
>
> Thank you for this feedback that helps us strengthen our work. We **evaluated the quality of the GPT-4 generated dataset with four biology subject matter experts (SMEs)** and established that our dataset is of high quality. We selected a subset of 60 main problems and 209 subproblems (an example can be found in Table 1) and used the questions provided below as the evaluation metrics. We present the percentage of “Yes” as the evaluation results.
> - For each main problem (60 in total), we used the following questions as the measurements and received perfect performance:
>     - Is the solution to the main problem factually correct? (Yes / No): 100%
>     - Does the subproblem represent key aspects of the main problem? (Yes / No): 100%
> - For each subproblem (209 in total), we use the following questions to evaluate each component:
>     - Is the answer to the subproblem factually correct? (Yes / No): 98.5%
>     - Is the hint helpful? (Yes / No): 96.2%
>     - Is the incorrect response relevant to the subproblem? (Yes / No): 97.6%
>     - Is the incorrect response really incorrect? (Yes / No): 97.6%
>     - Does the feedback successfully address the incorrect response? (Yes / No): 99.0%
>     - Is the subproblem related to the main problem? (Yes / No): 100%
>
> *Comment #2. Concerns about sending private data to third parties (GPT-4)*
>
> The synthetic conversation dataset is based on questions generated by GPT-4 in the scaffolding phase, and we **do not send any confidential information to GPT-4 for generation of either of our scaffolding and conversational datasets**. The conversation dataset can then be used to fine-tune an LLM such as the open-source LLaMA/ Vicuna, which can then be deployed privately. Once the models are trained, instructors and edtech companies can use the private models with some supervision to create datasets specific to their own problems. Additionally, creating the synthetic scaffolding and conversations datasets with GPT-4 is cost-effective, since it only costs approximately $50, whereas the creation of similar datasets by subject matter experts (SMEs) would require thousands of dollars.
>
> *Comment #3: Baselines to compare SPOCK against: (1) Without CLASS framework dataset fine-tuning, (2) Without textbook pre-training (3) Compare Vicuna-13B with other models*
>
> The primary objective of this work was to propose a feasible, generalizable, and scalable framework for leveraging LLMs to create ITS. We share findings from our preliminary investigations about these factors below:
> 1. We observed that the Vicuna-13B without the CLASS framework dataset fine-tuning does not follow the prompts and that the default behavior of the Vicuna model is to output the answer to the student question. This behavior does not support effective learning and may result in cheating behavior. We wanted to ensure that the bot nurtured deeper engagement with the learning content with probing questions.
> 2. Textbook pre-training impact: The MMLU [2] accuracy on “College Biology” and “High School Biology” subsets increased by 1.5% by textbook pre-training.
> 3. At the time of submission, the Vicuna models were the best open-source LLMs available. Since better models will emerge, the Vicuna model should be interpreted as a proof-of-concept that validates that our framework can transform a wide variety of LLMs into effective ITSs.
>
> *Question (a,b): Will the datasets and models be released?*
>
> Yes, the datasets, models, and code will be publicly available. They can be anonymously accessed here: https://anonymous.4open.science/r/Tutorbot-Spock-FB4F.
>
> *Question (c): Does L046-047 insinuate that existing ITS systems do not consider “learning science principles”?*
>
> From the sentence you mentioned, we **did not imply that current ITS do not take into account learning science principles**. We will rephrase this sentence to clearly communicate the same. We simply wanted to highlight that we are the first to develop an ITS that aligns the behavior of open-sourced LLMs to perform scaffolded learning.
>
> *Question (d): Is the claim at L141-143 about the capabilities of LLMs a bit general?*
>
> We will incorporate your advice to rephrase the sentence and make our claim about the “strong reasoning abilities” more specific to “question-answering abilities”.
>
> *Question (e): Does the working “like GPT-4” at L225-227 imply the use of other language models for dataset generation?*
>
> Our approach can be adapted to all generative LLMs that incorporate domain-specific (in this case, biology) knowledge. We are simply using GPT-4 as an example to illustrate the efficacy of our framework. However, to the best of our knowledge so far, we have found only GPT-4 to be strong enough to create powerful, dynamic, and factually correct conversations for biology education.
>
> *Question (f): Do you have any analysis showing that RAG ensures SPOCK’s correctness?*
>
> We tested our retrieval strategies on the MMLU benchmark [2] on “College Biology” and “High School Biology” subsets and found that they **increased its accuracy by 3% when only the problem field is used**. Using the subproblems fields (L376-381) for retrieval **increased the accuracy by a further 3.5%**. Our novel technique for retrieval based on an open-source OpenStax Biology textbook using subproblems in conjunction with the main problem has subsequently been validated by [1]. We will add these quantitative results to the paper to highlight the effectiveness of our retrieval technique since it was the primary reason for including the subproblem field in the tutorbot response.
>
> *Question (g): Why wasn’t the evaluation done with students, instead of biology SMEs?*
>
> For our preliminary evaluation, we wanted to evaluate the quality of the conversations generated by our ITS (such as factual correctness, relevance, etc.), which requires strong biology domain knowledge. This is the reason we evaluate the system with biology SMEs. We agree that evaluation with real students is critical for the design of ITS, and we have planned to start our student evaluation this fall semester.
>
> *References*
>
> [1] A. Radhakrishnan et al., “Question Decomposition Improves the Faithfulness of Model-Generated Reasoning.” arXiv, Jul. 25, 2023. Accessed: Aug. 28, 2023. [Online]. Available: http://arxiv.org/abs/2307.11768
>
> [2] D. Hendrycks et al., “Measuring Massive Multitask Language Understanding.” arXiv, Jan. 12, 2021. Accessed: Aug. 28, 2023. [Online]. Available: http://arxiv.org/abs/2009.03300

---

### Official Review · Reviewer_85Eg · 2023-08-11

**Soundness:** 3

**Excitement:**

3: Ambivalent: It has merits (e.g., it reports state-of-the-art results, the idea is nice), but there are key weaknesses (e.g., it describes incremental work), and it can significantly benefit from another round of revision. However, I won't object to accepting it if my co-reviewers champion it.

**Paper Topic And Main Contributions:**

This paper proposes a framework for ITS, develop a system called SPOCK based on the proposed framework and evaluated the system

**Reasons To Accept:**

This paper makes valuable contributions to AIED. The evaluation of the system is reasonable and the proposed method for reducing the effect of hallucinations of LLM have on the system is valuable. I also appreciate the fact that the system was designed with some core principles of the learning sciences in mind and that the system was thoroughly evaluated by domain experts.

**Reasons To Reject:**

The authors should make it clear and explicit what the novelty of this paper is relative to other existing tutoring dialogue-based ITS and constraint-based scaffolding modeling ITS. This work overlaps with existing ITS so the novelty has to be made very clear. I can see some novelty but it has to be explicit.

I don't see any contribution this paper makes to NLP to get accepted into EMNLP. This paper seems more suited for an artificial intelligence in education conference like EAAI, ITS or AIED. Even for an applied NLP paper, there still has to be some valuable contributions to NLP for a paper to be accepted into an NLP conference and I cannot find any NLP contributions.

**Reproducibility:**

4: Could mostly reproduce the results, but there may be some variation because of sample variance or minor variations in their interpretation of the protocol or method.

**Reviewer Confidence:**

5: Positive that my evaluation is correct. I read the paper very carefully and I am very familiar with related work.

**Typos Grammar Style And Presentation Improvements:**

The second/last paragraph of Section 3.4 is not clear to me. That paragraph needs to be rewritten so it is clear exactly what the message is. I also can't tell if it is a recommendation to the research community of it is part of the experiment design for this paper. Whichever it is, the paragraph needs to be rewritten.

---

> ### Author Rebuttal · Authors · 2023-08-29
>
> We thank you for your valuable feedback regarding our LLM-based Intelligent Tutoring System (ITS) design framework and your appreciation of our novel retrieval technique to reduce hallucinations and the application of learning science principles to our design.
>
> *Comment #1: The novelty of this paper relative to other existing ITS*
>
> Our work is distinct from the extant work on ITS because it proposes a generalizable and scalable design framework to create LLM-based ITSs. To the best of our knowledge, we are **the first to propose an open-source LLM-based ITS**, with models and datasets available for download (https://anonymous.4open.science/r/Tutorbot-Spock-FB4F). To train our LLMs, the framework aligns the models to exhibit scaffolding properties (using our scaffolding dataset) and then applies them in conversations (using our conversational dataset). This is in distinct contrast to the development process for existing ITSs, which rely on labor-intensive manual methods to develop scaffolding constraints.
>
> *Comment #2: Contributions to NLP*
>
> 1. Conversational data is what separates high quality chatbots from pre-trained language models, as shown by the Vicuna models [3] (90% ChatGPT quality), which fine-tuned LLaMA-1 on 70K user-shared ChatGPT conversations. However, conversational data is difficult to acquire. To mitigate this important issue we **introduced a novel NLP technique to curate prompts for generating structured conversational datasets**. In our work, we demonstrated that generating high-quality synthetic conversations is feasible. This technique can be applied to generate conversations to support chatbots in domains far beyond education, such as human resources, e-commerce and insurance. One can adapt our prompting framework to curate conversations based on the domain and need.
>
> 2. Existing LLM prompting strategies are not amenable to creating explainable solutions. Therefore, we created a novel prompt design for generating synthetic conversational datasets that incorporate **explainability and a quick feedback mechanism**. The explicit generation of thought, decision and action (L926-932) fields provides rationale to the bot’s final responses (whose generation is conditioned on these fields), which further enhances the explainability of bots. Moreover, the predefined multiple-choice decision and action space captured in the "decision" field enables a quick feedback mechanism to obtain the models performance without relying on examining the actual bot’s open-ended response. Given the general black-box nature of LLMs, these mechanisms can be valuable for manifesting the inner mechanisms and even biases within the model.
>
> 3. LLM hallucinations must be eliminated in critical, high-stakes applications like education. To this end, we devised a **novel NLP-based retrieval technique using subproblems that improves performance by reducing hallucinations compared to just retrieving the main problems** (L376-381). Our technique has subsequently been validated by [1].  We are explicitly highlighting this point in our manuscript as a key NLP contribution of our work. To provide some quantitative results, using this technique with passages from the open-source OpenStax Biology textbooks on the MMLU benchmark [2] on “College Biology” and “High School Biology” subsets increased the accuracy by 3% when only the problem field is used. Using the subproblems fields (L376-381) for retrieval further increased the accuracy by 3.5%. These are significant improvements.
>
> 4. We conducted additional **subject matter expert (SME) evaluation on the GPT-4 generated datasets** that further validates GPT-4’s capability to generate high-quality synthetic datasets. We selected a subset of 60 main problems and 209 subproblems (an example can be found in Table 1 in the paper) that were evenly distributed across four SMEs for evaluation. We use the questions provided below as the evaluation metrics and present the percentage of “Yes” as the evaluation results. Based on the evaluation results from our biology SMEs, we are confident that our synthetic datasets are of very high quality.
>    - For each main problem (60 in total), we used the following questions as the measurements and received perfect performance:
>         - Is the solution to the main problem factually correct? (Yes / No): 100%
>         - Does the subproblem represent key aspects of the main problem? (Yes / No): 100%
>    - The evaluation also established that the subproblems (209 in total)  were contextually relevant and accurate based on the following questions:
>         - Is the answer to the subproblem factually correct? (Yes / No): 98.5%
>         - Is the hint helpful? (Yes / No): 96.2%
>         - Is the incorrect response relevant to the subproblem? (Yes / No): 97.6%
>         - Is the incorrect response really incorrect? (Yes / No): 97.6%
>         - Does the feedback successfully address the incorrect response? (Yes / No): 99.0%
>         - Is the subproblem related to the main problem? (Yes / No): 100%
>
> *Reproducibility*
>
> Our code with detailed instructions and links to downloading our new open-source datasets and models can be anonymously accessed here: https://anonymous.4open.science/r/Tutorbot-Spock-FB4F.
>
> *References*
>
> [1] A. Radhakrishnan et al., “Question Decomposition Improves the Faithfulness of Model-Generated Reasoning.” arXiv, Jul. 25, 2023. Accessed: Aug. 28, 2023. [Online]. Available: http://arxiv.org/abs/2307.11768
>
> [2] D. Hendrycks et al., “Measuring Massive Multitask Language Understanding.” arXiv, Jan. 12, 2021. Accessed: Aug. 28, 2023. [Online]. Available: http://arxiv.org/abs/2009.03300
>
> [3] W.-L. Chiang et al., “Vicuna: An Open-Source Chatbot Impressing GPT-4 with 90%* ChatGPT Quality.” Mar. 2023. [Online]. Available: https://lmsys.org/blog/2023-03-30-vicuna/

---

### Meta-Review · Area_Chair_BpF6 · 2023-09-18

**Recommendation:** 4

**Metareview:**

This paper proposes an Intelligent Tutoring System (ITS) framework called CLASS, trains a proof-of-concept ITS called SPOCK that aims to teach students biology, defines an evaluation protocol, and provides a qualitative evaluation of the built system with 4 experts demonstrating positive outcomes.

All reviewers agree that the paper has **a number of merits**. Specifically:
1. *Motivation*: As reviewers 85Eg and g1rc point out, the paper makes valuable contributions to the AI in education domain and provides a new perspective on how to combine LLMs and ITS. Moreover, this work is grounded in the Learning Science principles.
2. *Datasets construction*: Reviewers c1QD and g1rc highlight that the construction of scaffolding and conversational datasets for ITS is a valuable contribution.
3. *Methodology*: The proposed framework and methodology is mentioned among the paper's strengths by all reviewers. Moreover, Reviewer g1rc points out that the paper provides an example use case of how to combine multiple LLMs to build an NLP-based application.
4. *Use of domain experts for evaluation*: Finally, all reviewers also agree that qualitative evaluation using domain experts adds to the strengths of this submission.

At the same time, all reviewers have identified **several weaknesses** of this paper and areas for possible improvement, and posed a number of questions for the authors. The major weaknesses include:
1. *Novelty*: Reviewer 85Eg points out that, while this paper makes a contribution to AI in education, it may lack in novelty when considered in the context of NLP applications.
2. *Clarity of writing*: All reviewers have made suggestions on how the paper can be improved in terms of clarity of the points made. See their reviews and questions for more details.
3. *Lack of datasets' quality evaluation*: This weakness is highlighted by Reviewer c1QD. In addition, Reviewer g1rc expresses further concerns regarding the evaluation approach taken by the authors.
4. *Further limitations in the datasets construction*: As Reviewer c1QD points out, the framework dataset construction process was only performed using GPT-4 – a closed, commercial system with little transparency. As the reviewer proposes, testing the framework construction process using models beyond GPT-4 would ensure generalizability of the approach and provide a concrete, reproducible method for educators to build without using 3rd party models.
5. *Lack of meaningful comparison and ablation studies*: Reviewer c1QD also points out that no baselines to compare SPOCK against were used in this work. Moreover, the reviewer suggests running some ablation studies.

The authors did a thorough job addressing reviewers' questions. They have also acknowledged reviewers' suggestions on how the paper can be further improved. These changes should be integrated in the revised version of the paper.

---

### Decision · Program_Chairs · 2023-10-07

**Decision:**

Accept-Findings

**Comment:**

This paper proposes an Intelligent Tutoring System (ITS) framework called CLASS, trains a proof-of-concept ITS called SPOCK that aims to teach students biology, defines an evaluation protocol, and provides a qualitative evaluation of the built system with 4 experts demonstrating positive outcomes.

All reviewers agree that the paper has **a number of merits**. Specifically:
1. *Motivation*: As reviewers 85Eg and g1rc point out, the paper makes valuable contributions to the AI in education domain and provides a new perspective on how to combine LLMs and ITS. Moreover, this work is grounded in the Learning Science principles.
2. *Datasets construction*: Reviewers c1QD and g1rc highlight that the construction of scaffolding and conversational datasets for ITS is a valuable contribution.
3. *Methodology*: The proposed framework and methodology is mentioned among the paper's strengths by all reviewers. Moreover, Reviewer g1rc points out that the paper provides an example use case of how to combine multiple LLMs to build an NLP-based application.
4. *Use of domain experts for evaluation*: Finally, all reviewers also agree that qualitative evaluation using domain experts adds to the strengths of this submission.

At the same time, all reviewers have identified **several weaknesses** of this paper and areas for possible improvement, and posed a number of questions for the authors. The major weaknesses include:
1. *Novelty*: Reviewer 85Eg points out that, while this paper makes a contribution to AI in education, it may lack in novelty when considered in the context of NLP applications.
2. *Clarity of writing*: All reviewers have made suggestions on how the paper can be improved in terms of clarity of the points made. See their reviews and questions for more details.
3. *Lack of datasets' quality evaluation*: This weakness is highlighted by Reviewer c1QD. In addition, Reviewer g1rc expresses further concerns regarding the evaluation approach taken by the authors.
4. *Further limitations in the datasets construction*: As Reviewer c1QD points out, the framework dataset construction process was only performed using GPT-4 – a closed, commercial system with little transparency. As the reviewer proposes, testing the framework construction process using models beyond GPT-4 would ensure generalizability of the approach and provide a concrete, reproducible method for educators to build without using 3rd party models.
5. *Lack of meaningful comparison and ablation studies*: Reviewer c1QD also points out that no baselines to compare SPOCK against were used in this work. Moreover, the reviewer suggests running some ablation studies.

The authors did a thorough job addressing reviewers' questions. They have also acknowledged reviewers' suggestions on how the paper can be further improved. These changes should be integrated in the revised version of the paper.